# Influence of Overhanging Bleb on Corneal Higher-Order Aberrations after Trabeculectomy

**DOI:** 10.3390/jcm11010177

**Published:** 2021-12-29

**Authors:** Yu Mizuno, Kazuyuki Hirooka, Yoshiaki Kiuchi

**Affiliations:** Department of Ophthalmology and Visual Science, Hiroshima University, 1-2-3 Kasumi Minamiku, Hiroshima 734-8551, Japan; kazuyk@hiroshima-u.ac.jp (K.H.); ykiuchi@hiroshima-u.ac.jp (Y.K.)

**Keywords:** glaucoma, overhanging bleb, higher-order aberrations, trabeculectomy, mitomycin C

## Abstract

Recent advances in ocular aberrometry have revealed that ocular surgery increases ocular and corneal higher-order aberrations. This retrospective single-center study aimed to examine the effects of the overhanging bleb on corneal higher-order aberrations using a wavefront analyzer. We included 61 eyes from 50 patients with overhanging bleb after trabeculectomy with a fornix-based conjunctival flap using mitomycin C (overhanging bleb group) and 65 eyes from 54 glaucoma patients with no history of glaucoma surgery (control group). Corneal higher-order aberrations (total higher-order aberrations, coma aberrations, coma-like aberrations, spherical aberrations, and spherical-like aberrations) on a 4 mm pupil diameter were measured using the TOPCON KR-1W wavefront analyzer. Corneal coma aberrations were higher in the overhanging bleb group than in the control group (0.16 ± 0.13 μm and 0.10 ± 0.05 μm, respectively; *p* = 0.042). Corneal coma-like aberrations were also higher in the overhanging bleb group than in the control group (0.31 ± 0.32 μm and 0.16 ± 0.09 μm, respectively; *p* = 0.022). With an increasing ratio of cornea covered by the bleb to the entire cornea, all corneal higher-order aberrations increased except for corneal coma-like aberrations. Overhanging bleb after trabeculectomy with a fornix-based conjunctival flap using mitomycin C and its size influenced corneal higher-order aberrations.

## 1. Introduction

Glaucoma is an optic neuropathy characterized by gradual progressive morphological changes in the optic disc and visual field loss [1]. Trabeculectomy (TLE) is an effective surgical technique for lowering the intraocular pressure to slow the progression of visual field loss in glaucoma patients [2]. Antimetabolites in TLE, such as mitomycin C (MMC), have significantly improved the success rate of TLE. MMC inhibits the proliferation of fibroblasts, thereby preventing excessive postoperative scarring and enhancing the growth of the large bleb [3]. However, following TLE, patients occasionally complain of foreign body sensation, excessive tearing, sensitivity to light, and vision changes. Some of these patients may have overhanging blebs (OHBs), which are defined as oversized filtering blebs that cover part of the cornea and are caused by tear film instability [4]. Their incidence appears to be increasing with the introduction of antimetabolites [5,6]. In addition, several studies have shown that TLE results in changes in corneal keratometry and topography, and also astigmatism, which leads to a decline in visual acuity [7,8].

Recently, advances in ocular aberrometry have revealed that ocular surface disease or surgeries increase ocular and corneal higher-order aberrations (HOAs) [9,10]. Several studies have revealed that ocular surface diseases such as pterygium, the growth of conjunctival tissue covered in the cornea, affected the HOAs of the cornea [11,12,13]. Since OHB shares features with pterygium, we hypothesized that OHB might also be associated with corneal HOAs. However, changes in corneal HOAs in OHB after trabeculectomy with a fornix-based conjunctival flap using MMC have not yet been investigated. Here, we examined the effect of OHB on corneal HOAs.

## 2. Materials and Methods

In this retrospective, cross-sectional study, patients who attended the Department of Ophthalmology, Hiroshima University Hospital, Japan, were evaluated between April 2017 and November 2018. The study received approval from the institution’s ethics committee (E–-797), and the research adhered to the tenets of the Declaration of Helsinki.

Sixty-one eyes from 50 patients with OHB who had undergone TLE with a fornix-based conjunctival flap using MMC at least 3 months prior to entry were analyzed in this study (OHB group). The eyes with multiple glaucoma surgery were also included in the OHB group. Sixty-five control eyes from 54 glaucoma patients who had no history of prior surgical intervention, except uncomplicated cataract surgery at least 3 months prior to their entry, were concurrently recruited during a similar period (control group). Best-corrected visual acuity (BCVA) and intraocular pressure (IOP; Goldman Applanation Tonometer, Haag-Streit, Köniz, Switzerland) were measured. The anterior segment was observed using slit lamp microscopy. In addition, all eyes with IOLs were using monofocal lenses.

The TOPCON KR-1W wavefront analyzer (Tokyo, Japan) can calculate “corneal” HOAs from the shape of the cornea as well as HOAs of the entire eye. We used the TOPCON KR-1W wavefront analyzer to measure corneal HOAs for a 4 mm pupil diameter without dilating the pupil, and the data were expanded to Zernike polynomials. The magnitude was demonstrated as the mean root square (RMS). Based on our reports, we evaluated corneal wavefront aberrations for coma (C^−1^_3_ and C^1^_3_), spherical aberrations (C^0^_4_), the RMS of the third-order, fourth-order, and total HOAs. The RMS of the third-order Zernike coefficients (the square root of the sum of the squared coefficients of C^−3^_3,_ C^−1^_3_, C^1^_3_, and C^3^_3_) was considered a coma-like aberration. The RMS of the fourth-order Zernike coefficients (the square root of the sum of the squared coefficients of C^−4^_4_, C^−2^_4_, C^0^_4_, C^2^_4,_ and C^4^_4_) was considered a spherical-like aberration. Finally, the total of HOAs was defined as the RMS of the magnitudes for the third- to fourth-order aberrations. All patients had BCVA of ≥20/40, which was enough to allow fixation on the target of the wavefront analyzer. Aberrometry measurements were automatically measured three times.

For clinical photographic images, we used the TOPCON SL-D8Z slit lamp mounted camera (Tokyo, Japan) to evaluate the anterior ocular segment (×10) from 40 degrees on the temporal side using a diffuser by 50–75 Ws (Watt seconds). OHB was diagnosed if the cornea was covered with bleb under the slit lamp. The dimensional parameters in clinical photographic images were calculated using the NIH image J software (Image J, National Institute of Health, Bethesda, Maryland, USA). The entire corneal area and the area of the bleb over the cornea were measured each as pixels. The ratio of cornea covered by bleb, the ratio of cornea covered by bleb to the entire cornea was calculated as the ratio of the OHB area in the cornea relative to the entire corneal area (Figure 1). We calculated the ratio of the cornea covered by the bleb as follows.

The cornea covered by bleb area (pixels)/the entire cornea (pixels) × 100 (%)

The exclusion criteria were BCVA <20/40 and patients with any history of ocular surgery (other than uncomplicated cataract surgery) for the control group. Patients of the OHB group were not excluded for having had glaucoma surgery several times. Additionally, patients of corneal, conjunctival ocular disease observed on a slit-lamp microscopy were also excluded (e.g., pterygium, superficial punctate keratopathy, corneal opacity, and corneal erosion).

### Statistical Analysis

Data were entered in an Excel spreadsheet (Microsoft Corp. Redmond, WA, USA) and analyzed using JMP software (ver. 14, SAS, Inc., Cary, NC, USA). Measurement data were expressed as the mean ± standard deviation with a 95% confidence interval. Continuous data from the two groups were analyzed by an independent *t*-test, whereas discrete data were analyzed using Pearson’s chi-square test. The influence of age, IOP, and the ratio of bleb area on aberrations was analyzed using multiple linear regression. BCVA was converted into logMAR units for analysis. Differences were statistically significant when the *p*-value was <0.05.

## 3. Results

The study included 126 eyes from 104 patients who were eligible. The control group comprised 65 eyes with a mean age of 66.23 ± 19.32 years (range, 20–89 years). The OHB group, which included patients with at least one TLE with a fornix-based conjunctival flap using MMC, comprised 61 eyes with a mean age of 67.47 ± 11.11 years (range, 25–90 years). There was no age or gender difference between the two groups. The mean BCVA in logMAR was −0.0086 ± 0.14 and 0.16 ± 0.30 in the control and OHB groups, respectively (*p* < 0.0001). The mean IOP was 14.00 ± 3.66 and 11.60 ± 4.43 in the control and OHB groups, respectively (*p* = 0.0012), which was significantly lower in the OHB group. However, spherical equivalents were −2.25 ± 3.65 and −2.98 ± 2.87 in the control and OHB groups, respectively (*p* = 0.22), and there was no difference between the groups (Table 1).

There were 46 eyes with primary open-angle glaucoma (POAG), five eyes with exfoliation glaucoma, four eyes with primary angle-closure glaucoma (PACG), four eyes with secondary glaucoma (three eyes with uvetic glaucoma, one eye with rubeotic glaucoma), and two eyes with childhood glaucoma in the OHB group. TLE was the most common surgical procedure (86.89%), followed by TLE + PEA + IOL (TLE, phacoemulsification and aspiration, and intraocular lens implantation). The average number of TLE surgeries was 1.33 ± 0.85 times (range, 1–5 times); 51 eyes had experienced only one surgery (83.61%), but a few eyes needed several TLE surgeries. The average period from the last surgery to the examination was 3.18 ± 3.81 years (range 0.25–20.76 years) (Table 2). 

Table 3 shows the association of corneal aberrations in control and OHB groups. Corneal coma aberrations were statistically higher in the OHB group than in the control group (0.16 ± 0.13 μm and 0.10 ± 0.05 μm; *p* = 0.042). Corneal coma-like aberrations were also higher in the OHB group (0.31 ± 0.32 μm and 0.16 ± 0.09 μm; *p* = 0.022). However, corneal total HOAs, spherical aberrations, and spherical-like aberrations were not different between the control and OHB groups (0.26 ± 0.14 μm and 0.36 ± 0.40 μm; *p* = 0.47, 0.04 ± 0.62 μm and 0.04 ± 0.07 μm; *p* = 0.72, 0.09 ± 0.71 μm and 0.16 ± 0.21 μm; *p* = 0.11, respectively). Analyzed for eyes with multiple glaucoma surgeries in the OHB group, corneal coma aberrations, corneal coma-like aberrations, and spherical-like aberrations were higher in the OHB group (0.23 ± 0.20 μm and 0.10 ± 0.05 μm; *p* = 0.0016, 0.37 ± 0.23 μm and 0.16 ± 0.09 μm; *p* = 0.0006, 0.22 ± 0.24 μm and 0.09 ± 0.71 μm; *p* = 0.0013, respectively). However, corneal total HOAs and spherical aberrations were not different between the control and the eyes with multiple glaucoma surgeries in the OHB group (0.26 ± 0.14 μm and 0.55 ± 0.57; *p* = 0.054, 0.04 ± 0.62 μm and 0.06 ± 0.11 μm; *p* = 0.57, respectively).

We measured the range of infiltration area of the bleb to the cornea and entire cornea using image J. The average ratio of cornea covered by bleb was 6.20 ± 5.46% (range 0.59–31.96%). Figure 2 shows the relationship between the ratio of cornea covered by bleb and corneal HOAs. There was a positive correlation between the ratio of bleb area and corneal total HOAs, corneal coma aberrations, corneal spherical aberrations, and corneal spherical-like aberrations (r = 0.38; *p* = 0.0026, r = 0.39; *p* = 0.0018, r = 0.34; *p* = 0.0071, r = 0.30; *p* = 0.021, respectively). Analyzed for eyes with multiple glaucoma surgeries in the OHB group, there was a positive correlation between the ratio of cornea covered by bleb and corneal total HOAs, corneal coma aberrations, corneal spherical aberrations, corneal coma-like aberrations, and corneal spherical-like aberrations (r = 0.83; *p* = 0.0027, r = 0.78; *p* = 0.0083, r = 0.85; *p* = 0.0020, r = 0.75; *p* = 0.013, r = 0.83; *p* = 0.0030, respectively).

Univariate regression revealed a significant positive relationship between the ratio of cornea covered by bleb for corneal total HOAs, corneal coma aberrations, corneal spherical aberrations, and corneal spherical-like aberrations (β = 0.38; *p* = 0.0026, β = 0.39; *p* = 0.0018, β = 0.34; *p* = 0.0071, β = 0.30; *p* = 0.02, respectively) (Table 4). Number of TLE surgeries also showed a positive relationship between corneal total HOAs and corneal coma aberrations (β = 0.28; *p* = 0.027, β = 0.28; *p* = 0.026, respectively). Univariate regression analysis demonstrated that there was no relationship between IOP <8 mmHg and corneal total HOAs, corneal spherical aberrations, corneal coma aberrations, corneal spherical aberrations corneal coma-like aberrations, and corneal spherical-like aberrations (β = −0.54; *p* = 0.17, β = −0.31; *p* = 0.45, β = −0.16; *p* = 0.71, β = −0.48; *p* = 0.23, β = −0.03; *p* = 0.94, respectively). Analyzed for eyes with multiple glaucoma surgeries in the OHB group, univariate regression revealed a significant positive relationship between the ratio of cornea covered by bleb for corneal total HOAs, corneal spherical aberrations, corneal coma-like aberrations, and corneal spherical-like aberrations (β = 0.83; *p* = 0.0027, β = 0.85; *p* = 0.0020, β = 0.75; *p* = 0.0030, β = 0.83; *p* = 0.0027, respectively). Multivariate regression analysis demonstrated a significant relationship with the ratio of cornea covered by bleb for corneal total HOAs, corneal coma aberrations, corneal spherical aberrations, and corneal spherical-like aberrations (β = 0.37; *p* = 0.0034, β = 0.40; *p* = 0.0013, β = 0.34; *p* = 0.0084, β = 0.28; *p* = 0.03, respectively) (Table 5). Multivariate regression analysis demonstrated that there was no relationship between <8 mmHg and corneal total HOAs, corneal spherical aberrations, corneal coma aberrations, corneal spherical aberrations corneal coma-like aberrations, and corneal spherical-like aberrations (β = −0.51; *p* = 0.12, β = −0.45; *p* = 0.43, β = −0.18; *p* = 0.69, β = −0.33; *p* = 0.52, β = −0.0031; *p* = 0.99, respectively). Analyzed for eyes with multiple glaucoma surgeries in the OHB group, multivariate regression revealed a significant positive relationship between ratio of cornea covered by bleb for corneal total HOAs, corneal spherical aberrations, corneal coma aberrations, corneal spherical aberrations corneal coma-like aberrations, and corneal spherical-like aberrations (β = 1.01; *p* = 0.0099, β = 1.13; *p* = 0.0039, β = 0.57; *p* = 0.016, β = 1.08; *p* = 0.0095, β = 1.00; *p* = 0.0098, respectively). Univariate regression analysis and multivariate regression analysis demonstrated there was no relationship between corneal HOAs and IOP <8 mmHg. 

Figure 3 shows the relationship between the duration of time period after the last TLE and corneal HOAs. There was a positive correlation between the duration of time period after last TLE and corneal total HOAs, corneal coma aberrations, corneal spherical aberrations, and corneal spherical-like aberrations (r = 0.42; *p* = 0.0007, r = 0.57; *p* < 0.0001, r = 0.42; *p* = 0.0007, r = 0.33; *p* = 0.0089, respectively). Also, the duration of the time period after the last TLE demonstrated positive correlation with the ratio of cornea covered by bleb (r = 0.33; *p* = 0.0089). Analyzed for eyes with multiple glaucoma surgeries in the OHB group, there was a positive correlation between the duration of the time period after the last TLE and the ratio of the cornea covered by the bleb (r = 1.00; *p* < 0.0001).

## 4. Discussion

TLE is a standard surgery for uncontrolled glaucoma; patients may expect to control glaucoma progression by lowering the IOP. Additionally, OHBs are a rare complication after TLE, and the mechanism underlying their formation is complex. Several factors, such as gravity on the OHB, the action of the eyelid, scar hyperplasia, and excessive aqueous over-filtration, may contribute to the formation of OHBs [14,15]. Different therapeutic methods (dissection, neodymium YAG laser, and autologous blood injection and compression suture) have been used to deal with these problems [14,16,17,18]. Following TLE, patients with OHB sometimes complain about vision change and dysesthesia, dry eye, and excessive tearing [19]. However, the mechanism that changes vision has not yet been fully understood. Several studies have revealed that ocular surgeries, such as intraocular lens implantation and cataract extraction [20,21,22] and scleral buckling [23], lead to a strong effect on corneal or ocular HOAs. Several studies have demonstrated changes in the refractive state or HOAs before and early post-TLE surgery [7,8,24,25]. In addition, some of these studies revealed that corneal or ocular HOAs were changed for 1 week–1 month after TLE, but they had returned to normal levels by 3 months [7,8].

In the present study, consistent with previous reports, we revealed a relationship between OHB on corneal HOAs. Pterygium is a degenerative condition of the conjunctiva with the subconjunctival tissue invading the cornea by destroying superficial layers of stroma and Bowman’s membrane [26]. The histopathology of OHB revealed tight connections with corneal tissues or the corneoscleral limbus and multiloculated cystic, rather than simply leaning on the cornea, suggesting that OHB induces changes in the ocular surface [4,27]. Therefore, we consider that OHB, in which the conjunctiva invades the cornea, similar to the pterygium, may also influence corneal HOAs. We previously reported case series to demonstrate changes in the refractive state before and after resection of OHB [28]. In those cases, removal of OHB reduced the symptoms of dysesthesia or corneal HOAs. Several studies showed that pterygium is associated with wavefront aberrations, and excision of pterygium reduces wavefront aberrations, indicating amelioration in visual function [10,13]. Therefore, we assume that the excision of OHBs may reduce wavefront aberrations. 

OHB is usually located in the superior quadrant (at least in all our cases it was) and covered part of the cornea. In our study, the analysis demonstrated that corneal coma aberrations and corneal coma-like aberrations were significantly higher in the OHB group than in the control group. Therefore, we believe that the result may support the assertion that OHB caused asymmetric optical distortion in the eyes.

The mechanism of OHB is not completely understood. Several factors, such as gravity on the OHB, the action of the eyelid, scar hyperplasia, and excessive aqueous over-filtration may contribute to the formation of OHBs [14,15]. These factors may cause the OHB to become larger over time. In this study, there was a positive correlation between the duration of time period after TLE and the ratio of corneal area encroached by OHB. Additionally, the duration of time period after TLE demonstrated a positive correlation with corneal total HOAs, corneal coma aberrations, corneal spherical aberrations, and corneal spherical-like aberrations. Based on our findings, the longer period after TLE may worsen OHB, and it seems to be correlated with exacerbation of corneal HOAs, except corneal coma-like aberrations.

During the first 3 months there are typically important changes in IOP, post-operative manipulations (e.g., injections, suture lysis) and important changes in bleb anatomy and configuration due to the scarring process. These changes as potential causes of the corneal aberrations have been reported in ocular or corneal aberrations, except spherical aberrations that were increased at 1–4 weeks after TLE, but they returned to control levels at 1–3 months [8,25]. The authors concluded that temporary ciliary body edema following TLE could change the thickness and position of the lens, and ACD induced the disturbance in ocular HOAs. However, changes of ocular HOAs returned by 3 months, suggesting that those changes in ciliary body edema may return to the control levels by 3 months after TLE surgeries. Because OHB is one of the late complications, in our study, all cases were analyzed 3 months or more after TLE. Like cataract surgery, as for small-incision cataract surgery, changes in corneal aberrations may occur early after surgery; however, these changes gradually returned to preoperative values by 2 or 3 months after surgery [29,30,31]. In our study, all cataract surgeries were performed with small incisions using a phacoemulsification platform, and all cataract surgery cases were measured 3 months or more after cataract surgery. Therefore, we considered that the effects of TLE surgery and cataract surgery themselves are negligible in our study. In the current study, the duration of the time period after the last TLE showed an association with corneal HOAs (except corneal coma-like aberrations) and the ratio of bleb to the cornea. It was speculated that increases in the duration may affect permanent stable changes in corneal HOAs after glaucoma surgery, especially in the OHB eyes. 

Therefore, although, the mechanism underlying OHBs formation is complex, for example, for preventing excessive aqueous over-filtration, it may be useful to ensure an adequate amount of aqueous humor to the conjunctiva, or if OHBs would once occur, it might be useful for resecting them earlier to prevent corneal HOAs, causing visual disturbance.

Our study has several limitations. First, it includes few participants and a lack of anterior chamber depth measurement (ACD). He [32] reported that corneal asphericity and ACD play important roles in determining peripheral wavefront aberrations. However, Jo et al. [33] suggested that the total HOAs change showed no correlation with ACD change before and after TLE using MMC. Further research on ACD and corneal aberrations is required. Second, the tear film fluctuation and OHB heights were not measured in our study. Numerous studies [34,35,36] have shown that the tear film instability led to wavefront HOA changes, and Ji et al. [19] reported that TLE, especially bleb height, was related to ocular surface instability. OHB may cause ocular surface instability to corneal aberrations. In our study, we excluded from the OHB group patients with corneal damage observed with slit-lamp microscopy, but did not fully investigate whether there was a dry eye or excessive tears such as tear film instability and the effect of eye drops. Further research to study the correlation between ocular stability and corneal aberrations in OHB eyes must address this problem. Third, in this study, we did not assess change of corneal HOAs or symptoms of dysesthesia before and after resection of OHB. Previously, we reported a case series in which surgical removal of OHB reduced the corneal HOAs and symptoms of dysesthesia [28]. However, in order to provide useful information about the benefit of resecting OHB, further research to evaluate the relation between corneal HOAs and symptoms before and after resection of OHB is needed.

In conclusion, OHBs after TLE with a fornix-based conjunctival flap using MMC increased corneal coma aberrations and coma-like aberrations. The ratio of the cornea covered by the bleb positively correlated with the duration of the time period after TLE and corneal HOAs, except for coma-like aberrations. We conclude that increases in the proportion of OHB in the cornea may worsen corneal HOAs, causing visual disturbances in the late period after TLE with a fornix-based conjunctival flap using MMC. 

## Figures and Tables

**Figure 1 jcm-11-00177-f001:**
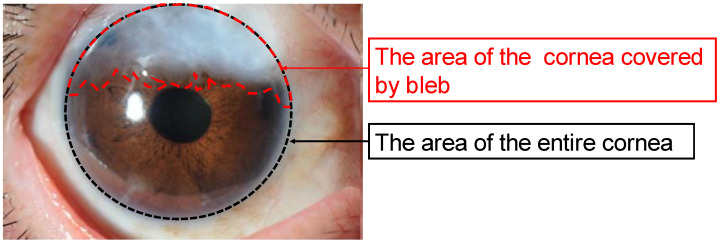
Clinical photographic image using slit lamp mounted camera and dimensions measured using NIH image J software. The ratio of cornea covered by bleb is the ratio of area of the cornea covered by bleb to entire cornea area. The red and black lines are the cornea covered by bleb area and entire cornea area, respectively.

**Figure 2 jcm-11-00177-f002:**
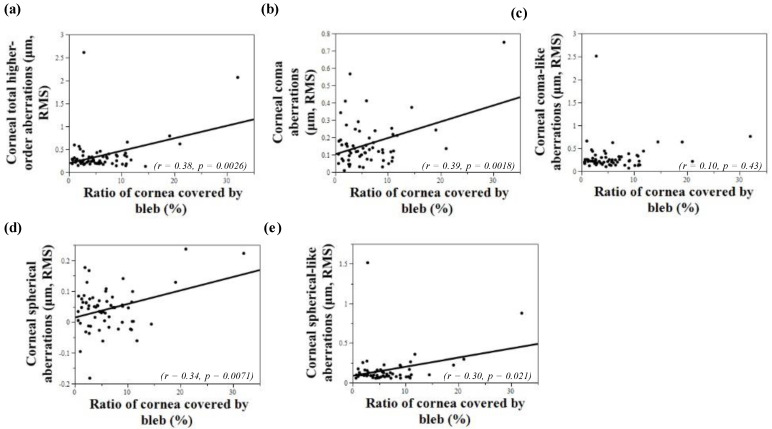
A linear regression comparison of the ratio of cornea covered by bleb with values of corneal higher-order aberrations (HOAs). (**a**) There is a positive correlation between the ratio of cornea covered by bleb and corneal total HOAs (r = 0.38; *p* = 0.0026). (**b**) There is a positive correlation between the ratio of cornea covered by bleb and corneal coma aberrations (r = 0.39; *p* = 0.0018). (**c**) Corneal coma-like aberrations showed no correlation between the ratio of cornea covered by bleb (r = 0.10; *p* = 0.43). (**d**) There is a positive correlation between the ratio of cornea covered by bleb and corneal spherical aberrations (r = 0.34; *p* = 0.0071). (**e**) There is a positive correlation between the ratio of cornea covered by bleb and corneal spherical-like aberrations (r = 0.30; *p* = 0.021).

**Figure 3 jcm-11-00177-f003:**
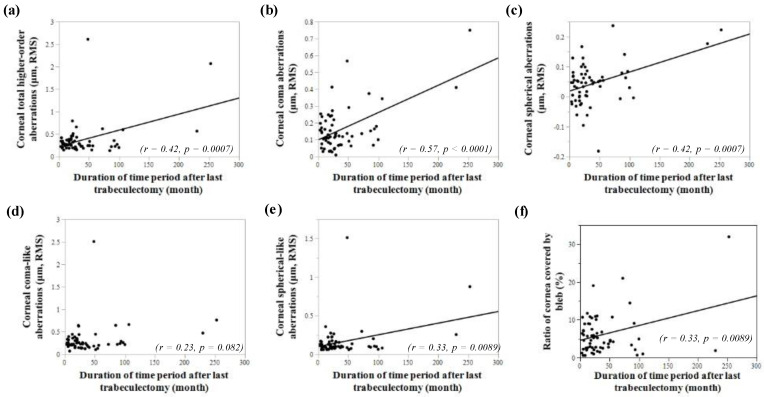
A linear regression comparison of the duration of time period after last TLE and corneal higher-order aberrations (HOAs). (**a**) There is a positive correlation between duration of time period after last TLE and corneal total HOAs (r = 0.42; *p* = 0.0007). (**b**) There is a positive correlation between duration of time period after last TLE and corneal coma aberrations (r = 0.57; *p* < 0.0001). (**c**) There is a positive correlation between duration of time period after the last TLE and corneal spherical aberrations (r = 0.42; *p* = 0.0007). (**d**) Corneal coma-like aberrations showed no correlation between duration of time period after the last TLE (r = 0.23; *p* = 0.0082). (**e**) There is a positive correlation between duration of time period after the last TLE and corneal spherical-like aberrations (r = 0.33; *p* = 0.0089). (**f**) There is a positive correlation between duration of time period after last TLE and the ratio of cornea covered by bleb (r = 0.33; *p* = 0.0089).

**Table 1 jcm-11-00177-t001:** Demographic data of participants included in the study.

	Control (*n* = 65)	OHB (*n* = 61)	*p*
Age (years)	66.23 ± 19.32	67.47 ± 11.11	0.66
Gender (Male/Female)	42/23	34/27	0.31
BCVA (logMAR)	−0.0086 ± 0.14	0.16 ± 0.30	<0.0001
IOP (mmHg)	14.00 ± 3.66	11.60 ± 4.43	0.0012
Lens status (phakic/IOL)	43/22	27/34	0.013
Spherical equivalents	−2.25 ± 3.65	−2.98 ± 2.87	0.22

OHB: overhanging bleb, BCVA: best-corrected visual acuity, IOP: intra ocular pressure.

**Table 2 jcm-11-00177-t002:** Clinical characteristics of control and overhanging bleb eyes.

	Control (*n* = 65)	OHB (*n* = 61)
Type of glaucoma		
PACG (%)	3 (4.6)	4 (6.6)
POAG (%)	37 (56.9)	46 (75.4)
Exfoliation G (%)	4 (6.2)	5 (8.2)
Uveitic G (%)	0 (0)	3 (4.9)
Rubeotic G (%)	1 (0)	1 (1.6)
Childhood G (%)	2 (3.1)	2 (3.3)
Steroid-induced G (%)	3 (3.1)	0 (0)
PPG (%)	16 (24.6)	0 (0)
PAC (%)	1 (1.5)	0 (0)
Operation (First time)		
TLE (%)	-	53 (86.89)
TLE + PEA + IOL (%)	-	6 (9.84)
Ex-PRESS (%)	-	2 (3.28)
Average number of TLE surgeries	-	1.33 ± 0.85
1st time (%)	-	51 (83.61)
2nd time (%)	-	4 (6.56)
3rd time or more (%)	-	6 (9.84)
Period after the last surgery (year)	-	3.18 ± 3.81

OHB: overhanging bleb, PACG: primary angle-closure glaucoma, POAG: primary open-angle glaucoma, PPG: preperimetric glaucoma, PAC: primary angle closure, G: glaucoma, TLE: trabeculectomy, PEA + IOL: phacoemulsification and aspiration + intraocular lens implantation.

**Table 3 jcm-11-00177-t003:** Corneal aberrations in control and overhanging bleb eyes.

	Control (*n* = 65)	OHB (*n* = 61)	*p*
Corneal total higher-order aberrations (μm, RMS)	0.26 ± 0.14	0.36 ± 0.40	0.47
Corneal coma aberrations (μm, RMS)	0.10 ± 0.05	0.16 ± 0.13	0.042
Corneal spherical aberrations (μm, RMS)	0.04 ± 0.62	0.04 ± 0.07	0.72
Corneal coma-like aberrations (μm, RMS)	0.16 ± 0.09	0.31 ± 0.32	0.022
Corneal spherical-like aberrations (μm, RMS)	0.09 ± 0.71	0.16 ± 0.21	0.11

OHB: overhanging bleb, Total higher-order: magnitude of the third to sixth order, coma-like: third-order Zernike coefficients, sphericcal-like: fourth-order Zernicke coefficients. The influence of BCVA, IOP and lens status on aberrations were analyzed using multiple linear regression.

**Table 4 jcm-11-00177-t004:** Univariate regression analysis of corneal higher-order aberrations with associated factors in overhanging bleb eyes.

	Corneal Total Higher-Order Aberrations (μm, RMS)		Corneal Coma Aberrations (μm, RMS)		Corneal Spherical Aberrations (μm, RMS)		Corneal Coma-Like Aberrations (μm, RMS)		Corneal Spherical-Like Aberrations (μm, RMS)	
	β	*p*	β	*p*	β	*p*	β	*p*	β	*p*
Ratio of cornea covered by bleb	0.38	0.0026	0.39	0.0018	0.34	0.0071	0.10	0.43	0.30	0.021
Number of TLE ≥ 2	0.21	0.11	0.24	0.061	0.099	0.45	0.077	0.56	0.13	0.31
IOP < 8	−0.54	0.17	−0.31	0.45	−0.16	0.71	−0.48	0.23	−0.03	0.94
Age	v0.03	0.84	0.14	0.28	0.0067	0.96	−0.046	0.73	−0.12	0.36

IOP: intra ocular pressure, TLE: trabeculectomy.

**Table 5 jcm-11-00177-t005:** Multivariate regression analysis of corneal higher-order aberrations with associated factors in overhanging bleb eyes.

	Corneal Total Higher-Order Aberrations (μm, RMS)			Corneal Coma Aberrations (μm, RMS)			Corneal Spherical Aberrations (μm, RMS)			Corneal Coma-Like Aberrations (μm, RMS)			Corneal Spherical-Like Aberrations (μm, RMS)		
	β	*p*	VIF	β	*p*	VIF	β	*p*	VIF	β	*p*	VIF	β	*p*	VIF
Ratio of cornea covered by bleb	0.37	0.0034	1.01	0.40	0.0013	1.01	0.34	0.0084	1.01	0.093	0.48	1.01	0.28	0.03	1.01
Number of TLE ≥ 2	0.038	0.78	1.29	0.048	0.72	1.29	−0.085	0.55	1.29	0.038	0.80	1.29	0.0035	0.98	1.29
IOP < 8	−0.51	0.12	1.25	−0.45	0.43	1.25	−0.18	0.69	1.25	−0.33	0.52	1.25	−0.0031	0.99	1.25
Age	−0.025	0.84	1.02	0.20	0.11	1.02	0.0056	0.65	1.02	−0.024	0.86	1.02	−0.072	0.57	1.02

IOP: intra ocular pressure, TLE: trabeculectomy, VIF: varianceinflation factor.

## Data Availability

The data analyzed in this study are available from the corresponding author on reasonable request.

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
