# Peer review of "Influence of Overhanging Bleb on Corneal Higher-Order Aberrations after Trabeculectomy"

_jcm, 2021, doi:10.3390/jcm11010177_

Round 1

Reviewer 1 Report

An overriding feature of overhanging blebs clinically is dysaesthesia - described as FB sensation due to tear film instability. This latter feature is known to influence corneal higher order aberrations and therefore the non-inclusion of ocular surface assessments in this regard is one of the major limitations of the research methodology.

Inclusion of patients who underwent phaco in both groups without mentioning what type of IOL these eyes received (aspheric or not, monofocal, multifocal etc) can also influence HOAs.

Lens status can also influence HOAs. The two groups had significant difference in phakic/IOL status. The eyes that were phakic, did they have cataract? And to what extent? It is a known fact that in eyes with mild nuclear cataracts the spherical-like aberration is dominant whereas in cortical cataracts it is coma-like aberration. (ref: Sachdev N, Ormonde SE, Sherwin T, McGhee CN. Higher-order aberrations of lenticular opacities. J Cataract Refract Surg. 2004 Aug;30(8):1642-8.)

Furthermore drawing parallels with pterygium may also be an oversimplification - there is enough evidence that it disintegrates the Bowman's layer and can invade the stroma too. (ref: Cameron ME Histology of pterygium: an electron microscopic study.

Reviewer 2 Report

Please review, modify the manuscript and give answer to the following comments:

METHODS

  • Page 2, line 55. Please define overhanging bleb. Minimum size? Minimum invasion of corneal invasion in mm?
  • Page 2, line 79. Please consider changing the concept “ratio of bleb area” which seems a bit confusing to something different. Maybe….”Ratio of corneal are covered by the bleb” “Cornea covered by bleb” or other.
  • Page 2, line 80. Please define here how you calculated the parameter referred in previous point.
  • Page 3. Line 89. Please confirm and clarify in the text that eyes with multiple glaucoma surgeries could be included. And if so, this group of cases should be analyzed a bit deeper because several filtration surgeries have undoubtedly more chances of changing corneal surgery that just one.

RESULTS

  • Page 3. Have authors evaluated if low pressures (under 10 mmHg, or even under 8 mmHg) increase the risk of corneal aberrations? Please describe.
  • Page 3, Table 2, heading. Should the third column have as tittle “OHB” and not TLE?

DISCUSION

  • Page 7, line 224. I am not sure that including cases only after 3 months of surgery is a limitation. In fact, I believe it is the way to go if the authors want to evaluate permanent stable changes in corneal aberrations after glaucoma surgery. During the first three months there are typically important changes in IOP, postoperative manipulations (injections, suture lysis….) and important changes in bleb anatomy and configuration due to scarring process. All these changes should be mentioned as potential causes of the corneal aberrations shown by other studies.
  • Page 7. A global comment of the clinical consequences of the study could also be added. The findings of the study shows that migration of the bleb onto the cornea may objectively affect quality of vision. This fact should stimulate glaucoma surgeons to pursue and use techniques that would decrease the risk or anterior bleb migration.

Reviewer 3 Report

The authors have conducted a good study, comparing the effects of the overhanging bleb following trabeculectomy, on higher-order corneal aberrations. The study helps us in understanding the symptoms experienced by patients following trabeculectomy. Since overhanging blebs worsen over time, curious to know if the authors looked at correlation between the duration of time period after trabeculectomy and the extent of corneal aberrations. Also, extending the study and assessing the aberrations after surgical correction of the OHB and relating this to alleviation of patient's symptoms, would provide useful information about the benefit of surgery, instead of assuming this based on studies following excision of pterygium. (sentence 196)
